# Biomimetic temporal self-assembly via fuel-driven controlled supramolecular polymerization

Ananya Mishra[1], Divya B. Korlepara[2], Mohit Kumar[1], Ankit Jain[1], Narendra Jonnalagadda[1], Karteek K. Bejagam[2], Sundaram Balasubramanian [2] & Subi J. George [1]

Temporal control of supramolecular assemblies to modulate the structural and transient characteristics of synthetic nanostructures is an active field of research within supramolecular chemistry. Molecular designs to attain temporal control have often taken inspiration from biological assemblies. One such assembly in Nature which has been studied extensively, for its well-defined structure and programmable self-assembly, is the ATP-driven seeded self-assembly of actin. Here we show, in a synthetic manifestation of actin self-assembly, an ATP-selective and ATP-fuelled, controlled supramolecular polymerization of a phosphate receptor functionalised monomer. It undergoes fuel-driven nucleation and seeded growth that provide length control and narrow dispersity of the resultant assemblies. Furthermore, coupling via ATP-hydrolysing enzymes yielded its transient characteristics. These results will usher investigations into synthetic analogues of important biological self-assembly motifs and will prove to be a significant advancement toward biomimetic temporally programmed materials.

---

[1] Supramolecular Chemistry Laboratory, New Chemistry Unit, Jawaharlal Nehru Centre for Advanced Scientific Research (JNCASR), Jakkur, Bangalore 560064, India. [2] Molecular Simulations Laboratory, Chemistry and Physics of Materials Unit, Jawaharlal Nehru Centre for Advanced Scientific Research (JNCASR), Jakkur, Bangalore 560064, India. Correspondence and requests for materials should be addressed to S.B. (email: bala@jncasr.ac.in) or to S.J.G. (email: george@jncasr.ac.in)

Biological systems use self-assembly of proteins, controlled via various molecular cues, as a functional motif[1]. For example, in cytoskeleton, which is responsible for crucial functions like cell motility and nutrient trafficking inside the cells, the ability to assemble and disassemble in a temporal manner ultimately decides the functional outcome[2]. These systems undergo nucleation-elongation polymerization in a temporal manner and also show seeded growth characteristics. Interestingly, all the self-assembling processes are controlled by small molecules, which act as fuel for the system such as adenosine triphosphate (ATP) and guanosine triphosphate (GTP). The cellular metabolic machinery can simply control the concentration of these molecular motifs that ultimately governs the function of these systems by spending molecular currencies as fuel[3,4]. The assembly of G-actin into filaments also accelerates the rate of ATP hydrolysis by a factor of 40,000 due to changed molecular interactions[5].

Mimicking these characteristics of temporally controlled growth as well as decay in synthetic supramolecular systems is very important to acquire control over the structure and function, thereby enabling us to design life-like active materials[6–10]. Based on an increased understanding into the mechanistic insights[11–16] and pathway complexity of supramolecular polymerization[17,18] over the last decade, there have been recent attempts to create artificial systems that undergo controlled growth via chain-[19] or living supramolecular polymerization processes[20,21]. An unprecedented control on the structure and dispersity of supramolecular polymers has thus been achieved, either via kinetic interplay of a dormant metastable state harnessed via an off-nucleation pathway[22–26] or from elegantly designed conformationally dormant monomers[19]. These studies provided a controlled supramolecular polymerization of small monomers compared to the well-studied living crystallization-driven self-assembly of kinetically stable polymeric assemblies[27–30]. However, these structures are distant to the biological approach, which is triggered externally by biological co-factors[31]. On the other hand, recent attempts on biomimetic fuel-driven temporal control on the transient assembly of one-dimensional (1-D) organic nanostructures[32–39], are still far from mimicking natural systems as they have not shown fuel-driven seeded growth relevant to biological scenarios[40] and hence lack structural control on dispersity and degree of polymerization on the resultant nanostructures.

A desirable scenario thus would be to have the best of both these worlds and have a fuel-driven controlled growth and disassembly in a temporal manner[40]. As a representative example, if one looks at the tread milling behaviour of actin, monomer subunits both associate and dissociate and this phenomenon is kinetically regulated by the rate of ATP hydrolysis. For example, actin monomers polymerizes kinetically in the presence of ATP in cells via a nucleation-elongation mechanism and also exhibits seeded characteristics, resulting in monodisperse self-assembled structures. In addition, it depolymerizes in a temporal manner when ATP hydrolyses to ADP.

In the present study, a synthetic system has been designed to undergo a fuel-driven temporal supramolecular polymerization with structural control. Herein, we report a nucleation-growth and seeded supramolecular polymerization of a carefully designed monomer, which can be temporally controlled by a specific biological co-factor, ATP. We have further elucidated the polymerization pathway by detailed spectroscopic and theoretical studies, deciphering a unique fuel-driven seeded growth with structural control over the dispersity and degree of polymerization. We have finally shown that the self-assembly can be rendered transient by temporally modulating consumption of the fuel using a hydrolytic enzyme. Thus fuel-driven seeded and temporal assembly presented here provides a general strategy toward controlled supramolecular polymerization.

## Results

**Molecular design**. In our previous reports, we have shown unique designs where chromophores functionalized with phosphate receptors like dipicolylethylenediamine-zinc complex (DPA–Zn)[41,42] self-assemble into dynamic supramolecular polymers on binding with ATP. More recently, we have used an "enzyme in tandem" approach to attain transient helical conformations and transient supramolecular polymerization of these systems under non-equilibrium conditions[43–45]. However, the growth kinetics of these DPA–Zn functionalized monomers were very fast, hindering the ability to control over their structural features. In an attempt to have a control over the kinetics of nucleation, we have carefully designed and synthesized a unique π-conjugated monomer, oligo(p-phenylenevinylene) derivative, functionalized with DPA–Zn phosphate receptors (1, Fig. 1a, Supplementary Methods). Unlike previous designs, this molecule, due to its distinctive acceptor–donor–acceptor (A–D–A) structure, exists in slip-stacked dormant state that can then be activated on selective interaction with ATP (vide infra).

**Fuel-driven nucleation elongation**. First, we investigated the ATP-binding driven self-assembly of 1 ($c = 2 \times 10^{-5}$ M) using various spectroscopic methods like ultraviolet–visible (UV-Vis) absorbance, fluorescence, and circular dichroism (CD). 1 exists in a molecularly dissolved state in $CH_3CN$, and forms short aggregates in HEPES/$CH_3CN$, 90/10, v/v (Supplementary Figs. 4 and 5). Interestingly, as ATP is introduced to this solution, a time-dependent self-assembly of 1 is triggered, as elucidated by the changes in its spectroscopic signatures. Spectroscopic changes include the gradual redshift of the absorbance from 448 to 478 nm (Fig. 2a and Supplementary Fig. 6) and the appearance of a positive bisignated CD signal with positive and negative maxima at 525 nm and 434 nm, respectively, through an isodichroic point at 465 nm (Fig. 2b). These observations hint toward a temporal growth driven by molecular cue, ATP in our case. Kinetic spectroscopic changes categorically indicate ATP-binding induced J-type self-assembly and left-handed helical organization of 1. Detailed titration studies further showed that 0.9 equiv. of ATP is required to saturate all the DPA receptor sites (Supplementary Fig. 7). Hence, all further experiments were performed with the same equiv. of ATP. Transmission electron microscopy (TEM) (Fig. 2c and Supplementary Fig. 8) and atomic force microscopy (AFM) (Supplementary Fig. 9) further confirmed the ATP-driven assembly of 1 to 1-D fibrous structures. Interestingly, TEM images of 1 with 0.9 equiv. of ATP, stained with uranyl acetate (1 wt% in water) showed uniform striations of 3.5 nm, which closely matches the width of the ATP-bound 1 stacks (Supplementary Fig. 10), suggesting that fibres in TEM represent molecular 1-D fibres.

To get further insights into the growth mechanism, we probed the temporal process by various experimental techniques with the hindsight of probing hierarchical stages of self-assembly. Time-dependent kinetics (monitored at 500 nm) showed the presence of a CD silent state of about 280 s (red plot, Fig. 2d). After this time lag, the CD signal increased sharply with time in a nonlinear manner indicating a cooperative self-assembly, which finally saturates at around 1000 s. This time-delayed self-assembly was also reflected in the absorbance (blue plot, Fig. 2d). But interestingly, the time lag observed in absorbance ($t_{lag}(Abs_{500 nm})$ = 203 s) was lesser than the CD time lag ($t_{lag}(CD_{500 nm})$ = 280 s)[46] (Supplementary Fig. 11 and Supplementary Table 1). This is indicative of the fact that before the emergence of chiral reorganization in the assembly, there is a significant growth of

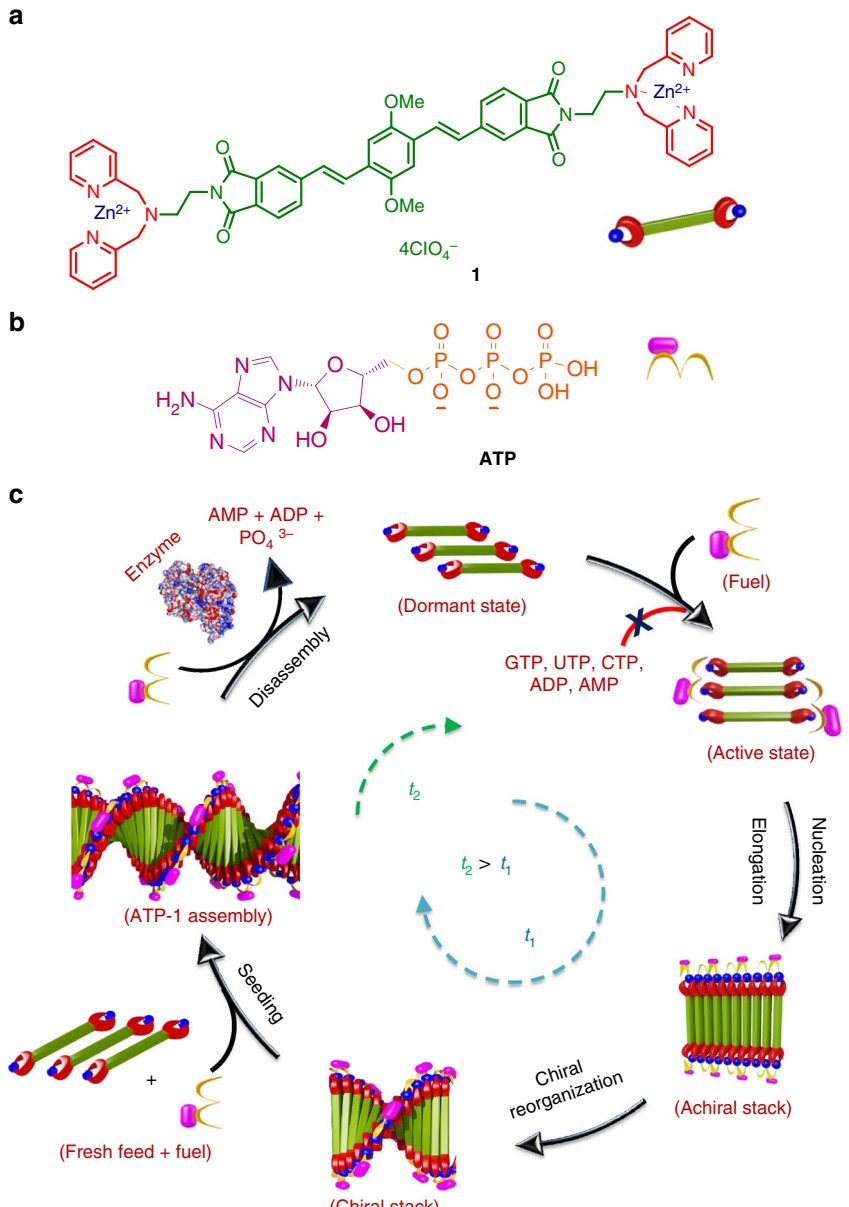

**Fig. 1** ATP-driven temporal assembly of **1**. **a, b** Molecular structures of **1** and ATP, respectively. **c** Schematic representation of ATP-selective nucleation-growth and seeded assembly of **1** with enzymatically modulated transient characteristics. $t_1$ corresponds to the time required for the growth process and $t_2$ is the time required for disassembly on enzymatic hydrolysis of ATP where $t_2 > t_1$, rendering a transient assembly

the stack, in an achiral arrangement. These observations were further supported by detailed DLS studies, which showed a similar growth pattern in scattering intensity (kcps) (green plot, Fig. 2d) as well as size (magenta plot, Fig. 2d), with a $t_{lag}$ (~200 s) matching with that of $t_{lag}$(Abs$_{500\,nm}$). This clearly suggests that the time-dependent changes in absorbance at 500 nm indeed reflect the ATP-fuelled growth of **1**. Interestingly, the growth kinetics monitored with absorbance could be well-fitted to a two-step autocatalytic model, first proposed by Watzky and Finke[1], which has been described for the nucleation growth for various protein polymerization to get nucleation ($k_n$) and elongation rates ($k_e$). These observations prove that the ATP-driven self-assembly of **1** follows a nucleation-elongation mechanism ubiquitous in biological scenarios. With increasing temperature and increasing equiv. of ATP (Supplementary Fig. 13), an increase in $k_n$ as well as $k_e$ values was noticed (Supplementary Figs. 14 and 15, Supplementary Tables 2 and 3). Decrease in the values of $t_{lag}$ and $t_{50}$ (half-time, i.e.,

time required for completion of 50% of the process) (Supplementary Fig. 16) was also observed.

Furthermore, important mechanistic insights into the origin of ATP-driven nucleation-growth assembly were provided by molecular mechanics (MM) / molecular dynamics (MD) simulations along with the time-dependent changes in fluorescence. **1** has a unique A–D–A electronic structure where the central dimethoxy phenyl ring acts as the donor and the phthallic imides are the acceptor moieties[47,48]. As a result, in HEPES/CH$_3$CN (90/10, v/v), these molecules exist in a pre-associated slip-stacked conformation due to intermolecular charge-transfer interactions as evident from the emission and excitation spectra (Supplementary Fig. 4). Due to this pre-organized, slip-stacked native state of **1**, an allosteric re-arrangement of monomers, where they have to slide in plane, would be essential to re-enforce the binding with the ATP molecules. The sigmoidal increase in fluorescence with a time lag of 88 s indeed provided a definitive proof for such a time-

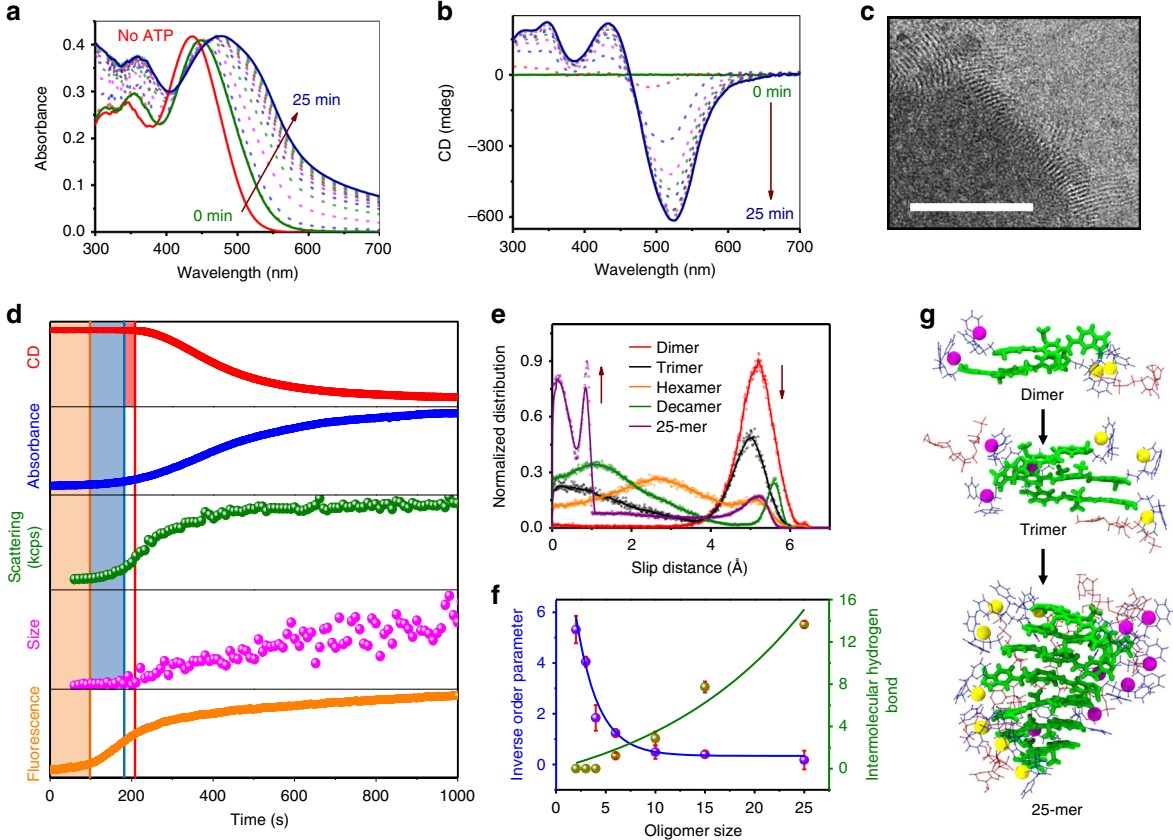

**Fig. 2** ATP-driven nucleation-growth self-assembly of **1**. **a** UV–Vis absorbance and **b** CD spectra represent time lapse spectra of ATP-driven growth of **1**. **c** TEM showing one-dimensional assemblies of **1** (scale bar = 100 nm). **d** Comparison of growth kinetics and lag phases of ATP-driven nucleation growth of **1** monitored by CD (mdeg), absorbance, scattering in kcps (kilo counts per second), size (nm) obtained from dynamic light scattering (DLS) and fluorescence. Yellow shaded area corresponds to time required for initial allosteric binding to commence; blue shaded area corresponds to time required for formation of stacks, triggered chronologically after allosteric binding of ATP to **1**, and red shaded area corresponds to lag phase after which helical ordering in stacks begins as elucidated by CD. Absorbance, CD, and emission were monitored at 500 nm ($c = 2 \times 10^{-5}$ M, HEPES/CH$_3$CN, 90/10, v/v, 0.9 equiv. ATP, 30 °C). **e** Slip distance distribution and **f** inverse order parameter (blue curve) and intermolecular H-bonds (green curve) per ATP for various oligomers extracted from MM/MD simulations of ATP-**1** stacks. The error bars were calculated using the standard error formalism, by considering the RMSD data over time as blocks. **g** Snapshots of MM/MD simulations of ATP-**1** oligomers showing decrease in slip distance with extent of polymerization (OPV core depicted by thick green sticks, DPA and ATP by blue and red sticks and Zn atoms as balls, with yellow on one side and magenta on the opposite, to show chirality of the assembly)

dependent allosteric-binding mechanism (orange plot, Fig. 2d), as shown by Hamachi and colleagues in pre-organized DPA receptors on peptide backbones (Supplementary Fig. 17)[49,50]. In agreement with these observations, MM/MD simulations showed that the slip distance between adjacently placed **1** decreases from 5.2 Å for a dimer (slip-stacked) to 1.35 Å for a 25-mer as allosteric binding of ATP kicks in (Fig. 2e, g and Supplementary Fig. 18). From Fig. 2e, it is clear that the normalized distribution of slip distance varies rapidly with oligomer size and the position of the peak shifts toward lower values with increasing oligomer size.

Thus, the allosteric-binding-induced supramolecular reorganization converts the inactive conformation of the monomers to an active conformation, which is the first chronological event en route to the nucleation process. Most importantly, the fluorescence increase is almost 70% over by the time changes in absorbance, characteristic of elongation, sets in. We envisage that the dormant self-assembled state in the present case is a thermodynamically stable one, which gets activated in the presence of fuel, hence different from previous kinetically trapped dormant assemblies[22–26]. MM/MD simulations were performed on preformed oligomers of various sizes with an initial twist angle 25°. As time progresses, the molecules slip over each other. The shorter oligomers show a huge slip as shown in Fig. 2e. To

understand the structural evolution of oligomers with time, we defined an inverse order parameter (IOP) as

$$\text{IOP}(n) = \text{RMSD}/(n - 1) \times \pi - \pi \text{ distance}, \quad (1)$$

where $n$ is the size of oligomer, RMSD is the root mean square deviation and the $\pi$–$\pi$ distance was chosen to be 3.8 Å. The fluxionality of the oligomers is estimated using RMSD, which is calculated with reference to a configuration, chosen arbitrarily, from a well-equilibrated run. As longer oligomers exhibit a well-defined twist between the molecules, the magnitude of RMSD gives an estimate of helicity, more the deviation less the helicity. In order to obtain a dimensionless quantity from RMSD, the same was divided by the mean $\pi$–$\pi$ distance of the 25-mer (i.e., 3.8 Å) and the number of consecutive pairs in an assembly, which is $(n - 1)$. From (Fig. 2f, blue curve and Supplementary Fig. 19), the increased stability of the assemblies as the oligomer size increases from a dimer to a 25-mer, is evident from the decrease in IOP. RMSD itself shows a weak decreasing dependence on oligomer size (Supplementary Fig. 20). Remarkably, this is in line with the increase in number of intermolecular H-bonds per ATP with the increase in oligomer size, seen in the simulated assemblies (Fig. 2f, green curve). Hence, we envision that this

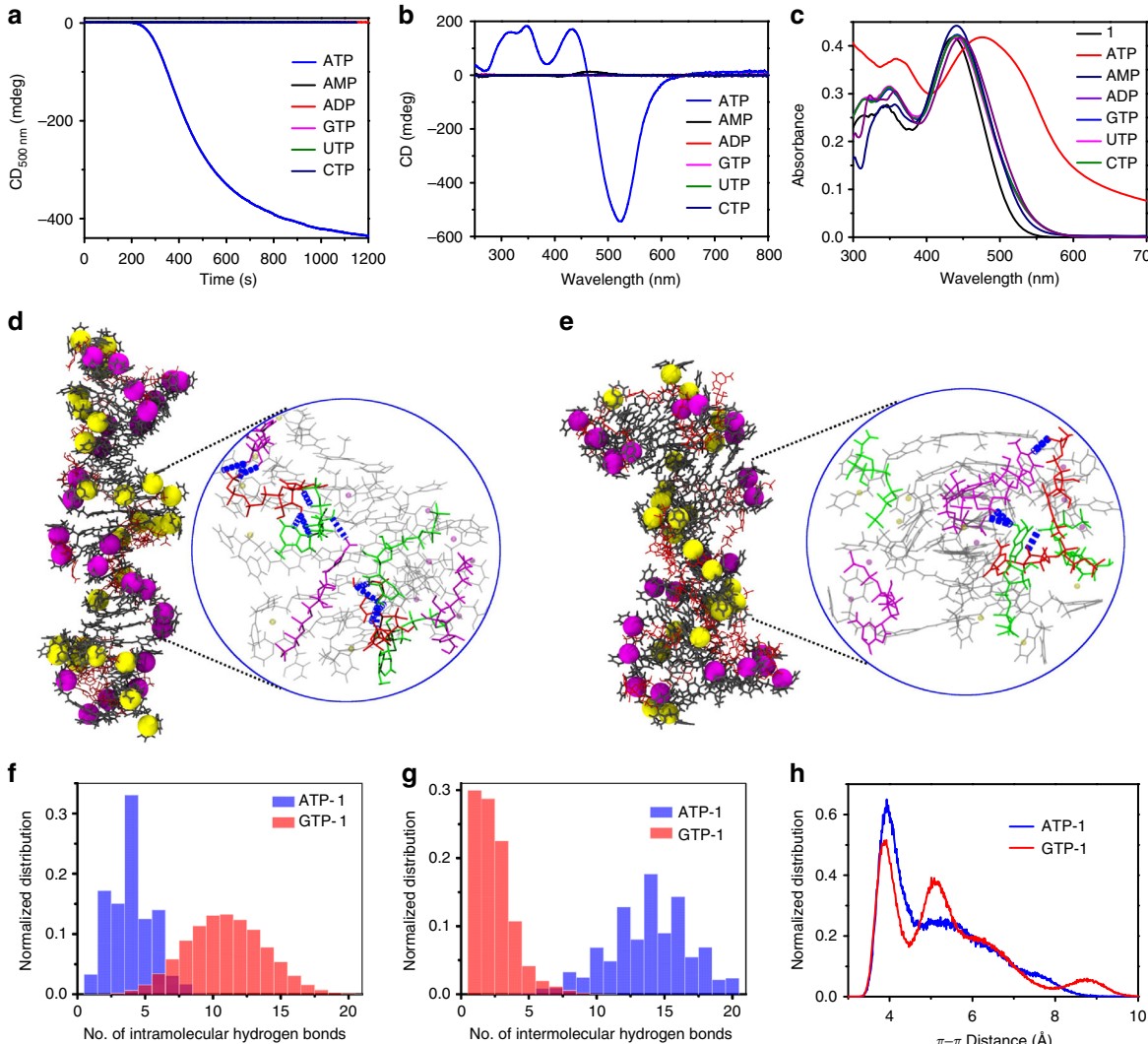

**Fig. 3** ATP-selective nucleation-growth of **1**. **a** CD intensity, **b** CD spectra, and **c** absorbance spectra of **1** with various phosphates ($c = 2 \times 10^{-5}$ M, HEPES/ CH$_3$CN, 90/10, v/v, 0.9 equiv. ATP, AMP, ADP, GTP, UTP and CTP, 30 °C). Final snapshots of **d**, ATP and **e**, GTP-bound stacks of **1** (ATP-**1** and GTP-**1**) obtained from MM/MD simulations. Zoomed images in **d** and **e** show intermolecular H-bonding between the triphosphates. Zinc atoms have been depicted as balls of two different colours (magenta and yellow) for easy identification of the helicity in the oligomer. Molecules of **1** are depicted as black sticks (in **d** and **e**), while ATP and GTP are in red. In zoomed images of **d** and **e**, intermolecular H-bonds are shown as dashed light blue lines. Each ATP/ GTP molecule is depicted in a distinct colour to make it obvious that the hydrogen bonds are intermolecular in character. Comparison between **f**, intramolecular H-bonds, **g**, intermolecular H-bonds and **h**, $\pi-\pi$ distance between two adjacent molecules of **1** in ATP-**1** and GTP-**1** simulated assemblies. The peripheral molecules in the stack are relatively more flexible, which leads to a higher $\pi-\pi$ distance, giving a bimodal signal in the case of ATP-**1** assembly. On the other hand, in the case of the GTP-**1** assembly, the molecules in stack are not well packed, which gives rise to a multimodal distribution

increased stability brought about by the presence of chiral adenosine group at the periphery of the assemblies could be the reason for induction of supramolecular chirality beyond a critical stack length. These explanations offer a mechanistic insight into the various hierarchical processes taking place during the ATP-driven nucleation-growth self-assembly of **1**.

**ATP-selective cooperative growth.** Remarkably, time-dependent spectroscopic studies with various phosphates have revealed that, other adenosine phosphates (ADP and AMP) and various tri-phosphates (GTP, UTP, and CTP), are unable to induce the growth of the stacks (Fig. 3a, b and Supplementary Fig. 21) though all of them bind to the monomers of **1** (Fig. 3c). This confirms that the nucleation elongation of **1** is completely selective to the fuel (ATP), which is a striking resemblance to the actin self-assembly. In order to get further insights into this unique fuel-selective nucleation-growth assembly of **1**, detailed

MM/MD simulations were carried out for ATP-**1** and GTP-**1** stacks, as ATP and GTP are the most biologically relevant tri-phosphates that trigger actin and microtubules self-assembly in Nature. Detailed simulations starting from a preformed oligomer containing 25 molecules (Supplementary Fig. 43) in explicit solvent at 298.15 K for 60 ns suggests that ATP-**1** stacks are more stabilized than the GTP-**1** stacks due to a larger number of intermolecular hydrogen bonds in the former (Fig. 3d, e). Although the mean number of intramolecular hydrogen bonds in GTP-**1** is more than that in ATP-**1** as seen from the normalized distributions (Fig. 3f), such hydrogen bonds do not contribute to the stack stability. In contrast, the average number of inter-molecular hydrogen bonds in ATP-**1** is much larger than that in GTP-**1** (13 and 2 respectively, see Fig. 3g). We envisage that intermolecular hydrogen bonds in ATP-bound stacks bring the molecules of **1** closer to each other to increase $\pi-\pi$ interaction and which stabilizes the stacks. Mean values of $\pi-\pi$ distances for

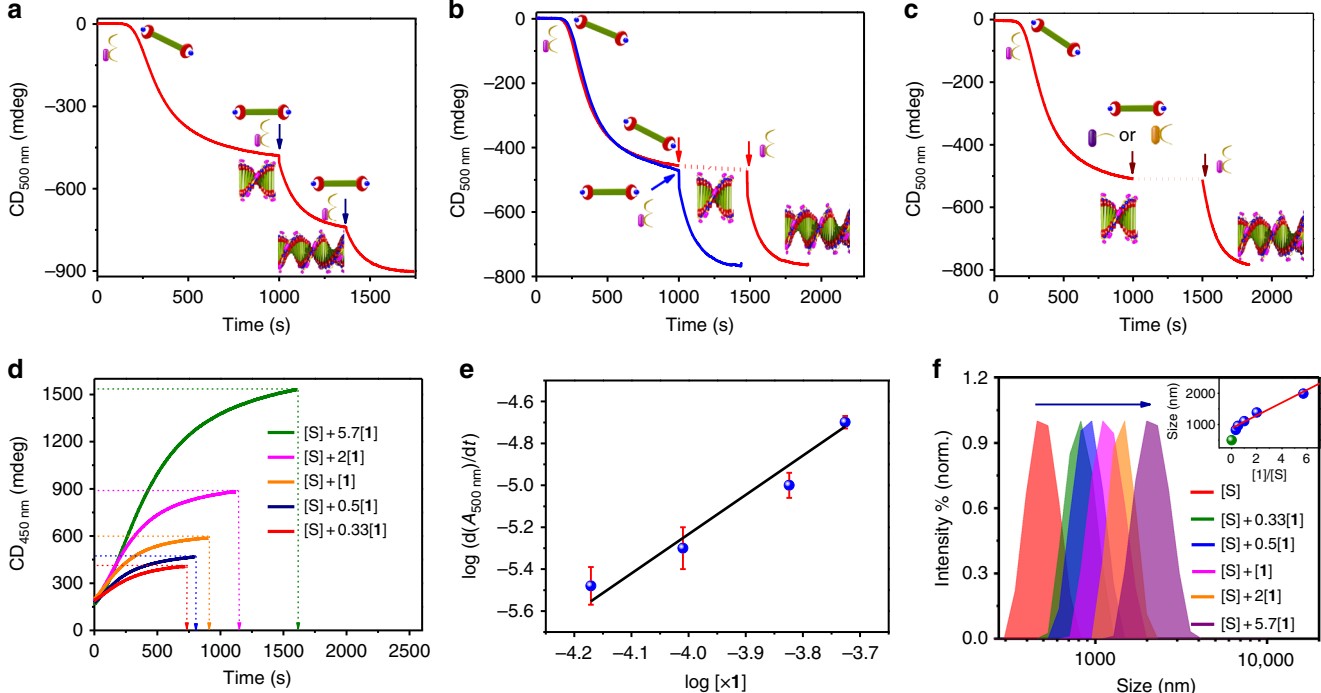

**Fig. 4** ATP-selective seeded supramolecular polymerization of **1**. **a** CD signal showing seeded supramolecular polymerization of **1** upon two subsequent addition of monomers with ATP ($c = 2 \times 10^{-5}$ M). **b** Overlaid spectra of CD changes showing the simultaneous (blue) and consecutive (red) addition of **1** and ATP to pre-grown seed. **c** CD signals showing no changes on addition of 5 equiv. of AMP or 2 equiv. of ADP along with **1** to a pre-grown seed, but increase in CD signal as soon as 0.9 equiv. of ATP was added to it. (Seed conc. [S] = $2 \times 10^{-5}$ M, fresh feed conc. [**1**] = $1 \times 10^{-5}$ M (5 μL of **1** ($5 \times 10^{-3}$ M stock) and 0.45 μL of ATP ($10^{-2}$ M)). **d** CD signal showing seeded polymerization for [**1**]/[S] ratios of 0.33:1, 0.5:1, 1:1, 2:1, and 5.7:1. **e** Log–log plot of the rate of change in absorbance as a function of [**1**] ($x = 2, 1, 0.5,$ and 0.33) exhibits a linear relationship with a correlation coefficient of 0.98. Error bars have been shown in red for data achieved from experiments carried out for a maximum of three times. **f** DLS data showing the increase in hydrodynamic radii from seed (S) to varying [**1**]/[S] ratios (inset shows the straight line fit of size increase for various seed concentrations where green ball is the seed size and blue balls are the sizes of seeded stacks) ([S] = $1 \times 10^{-5}$ M, [**1**] = $0.33 \times 10^{-5}$ M (1.65 μL), $0.5 \times 10^{-5}$ M (2.5 μL), $1 \times 10^{-5}$ M (5 μL), $2 \times 10^{-5}$ M (10 μL), $5.7 \times 10^{-5}$ M (2.85 μL) (HEPES/CH$_3$CN, 90/10, v/v, 0.9 equiv. ATP, 30 °C)

ATP-**1** and GTP-**1** stacks were found out to be 3.8 Å and 5.2 Å, respectively (Fig. 3h). The bimodal distribution of the $\pi-\pi$ distance is due to molecules present in the periphery of the stack, which are relatively more labile than the ones in its core. Hence, MD simulations strongly suggest that the propensity of intermolecular H-bonding in ATP-bound stacks than in GTP ones is responsible for the observed selectivity in the growth (vide supra).

**ATP-selective seeded supramolecular polymerization**. Having established the ATP-selective, fuel-driven cooperative self-assembly of **1**, we proceeded to investigate the seeded-assembly nature of these assemblies. A similar experiment was attempted by introducing a fresh feed of **1** along with 0.9 equiv. of ATP to a pre-grown ATP-**1** assembly (seed) and the CD (Fig. 4a and Supplementary Fig. 22) was monitored. The signal immediately started increasing with the absence of lag phase, which show that the fresh feed of monomers immediately grows on the stacks already present in the solution, instead of undergoing independent nucleation. Such a process could be repeated multiple times, again confirming the seeded supramolecular polymerization (Fig. 4a). Similar changes were also observed when absorption and emission changes were monitored on seeding (Supplementary Figs. 23 and 24). When the DLS data of seeded self-assembly was compared to the non-seeded stacks of the same net concentration, it was seen that the size of the seeded stack is larger than that of the non-seeded stacks reiterating the seeded nature of the growth (Supplementary Fig. 26). To prove the critical role of fuel, ATP, in the seeded self-assembly process, to a pre-grown stack of ATP-**1**, a fresh feed of **1** ($1 \times 10^{-5}$ M) was added without

ATP and the CD signal was monitored at 500 nm, which did not grow. However, when ATP is introduced again, the CD signal increased indicating the seeded growth (Fig. 4b and Supplementary Fig. 27). This further shows that the elongation of the stacks can be controlled with the concentration of fuel and a templated growth of the stack is required for the molecule to show seeded polymerization. In addition, other fuels like ADP and AMP could not trigger the seeded growth when introduced along with the fresh feed of monomers, in consistent with the ATP-selective cooperative growth (vide supra, Fig. 4c and Supplementary Fig. 28). However, subsequent addition of ATP to same solution could trigger the seeded growth again, as ATP could replace the bound guests through multivalent interactions. These observations are also backed by DLS measurements (Supplementary Fig. 29). This proves that seeding/growth can occur only in the presence of the ATP.

To gain insight into the kinetics of elongation process in seeded assembly, we added increased concentration of fresh feed of **1** along with appropriate amount of ATP to a fixed concentration of pre-grown seed ([S], $1 \times 10^{-5}$ M) and monitored the resulting changes in CD (Fig. 4d) and absorbance (Supplementary Fig. 30). With increasing concentration of monomer added (with increasing monomer to seed molar ratio, [**1**]/[S]), there was an increase in time taken by the seeded polymerization to saturate, evident from the $t_{50}$ and total time required for elongation observed (Supplementary Table 4), along with an increase in size seen from DLS (Fig. 4f and Supplementary Fig. 32), characteristic of a seeded self-assembly. Fitting the seeded growth kinetics to actin polymerization model given by Oosawa[51] and simplified by Moore[52], showed that

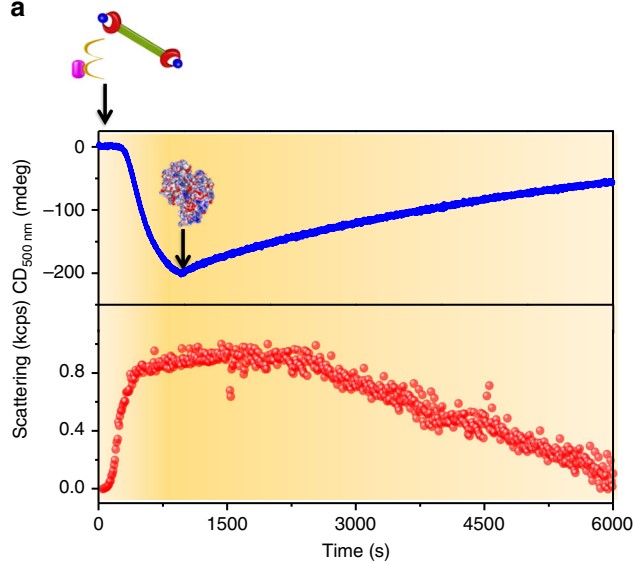

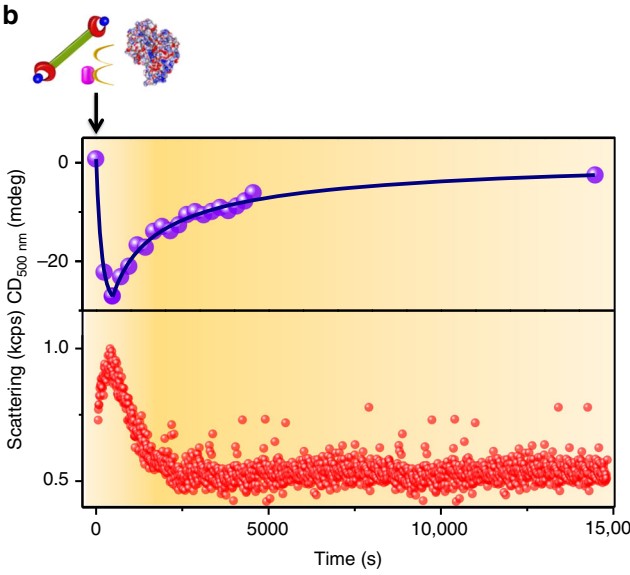

**Fig. 5** Enzyme mediated transient assembly of 1. **a** Passive assembly of ATP-**1** indicating growth followed by disassembly after apyrase is added into the elongated solution monitored using the changes in CD and scattering (0.9 equiv. ATP, 7.2 U apyrase), **b** active self-assembly of ATP-**1**, wherein the changes in CD and scattering indicates a transient self-assembly via temporal control over the ATP hydrolysis using apyrase (6.4 equiv. ATP, 0.6 U apyrase) ($c = 2 \times 10^{-5}$ M, HEPES/CH$_3$CN, 90/10, v/v, 30 °C)

the rate of elongation ($k_{e(seeded)}$) decreases with increasing [**1**]/[S] ratio (Supplementary Fig. 31 and Supplementary Table 5). In addition, the logarithm plot of apparent rate of growth calculated from the absorbance changes versus the concentration of added monomers gave a linear relation (Fig. 4e), indicating that the supramolecular polymerization reaction is of first order with respect to the fresh feed concentration, which concludes a chain growth-like seeded supramolecular polymerization. Remarkably, the plot of size increase (Inset of Fig. 4f) value from the DLS data against [**1**]/[S] showed a linear increase, characteristic of seeded supramolecular polymerization. This fuel-driven, seeded supramolecular polymerization strategy provided controlled polymer lengths, as evident from the low and consistent dispersity values (average value of 0.07) derived directly from DLS with good quality correlograms (Supplementary Figs. 33 and 34), analogous to the recently reported living supramolecular polymerization processes[22–26].

**Transient self-assembly**. Another important aspect of the temporal self-assembly of naturally occurring cytoskeleton proteins such as microtubules and actin filaments is their fuel-driven transient supramolecular polymerization under non-equilibrium. Since the fuel-driven growth of **1** is very selective to ATP, we envisage that temporally controlled enzymatic hydrolysis of ATP to ADP would bring about a transient assembly, analogous to actin. By using two complementary phosphoryl transferase enzymes, we have recently reported the transient helicity and transient assembly of a supramolecular polymer with no fuel-selectivity[43,44]. In the present study, we have chosen potato apyrase as the enzyme to perform depolymerization studies due to its higher activity even at lower concentrations and in the presence of CH$_3$CN[53,54]. Control experiments of pre-grown ATP-**1** with varying units of apyrase (Fig. 5a, Supplementary Fig. 35a) showed enzyme responsive disassembly of the stacks. The rate of disassembly increases with increasing units of apyrase hinting toward an enzymatic hydrolysis-driven mechanism of depolymerization. On hydrolysing ATP-**1** stacks in presence of 0.9 equiv. of ADP, absence of an initial lag in CD signal confirms that the hydrolysis of phosphates is not selective to unbound phosphates (Supplementary Fig. 35b). In order to extend this passive enzyme response of the stacks to the active regimes, we have standardized the concentrations of the fuels and enzymes in such a way that significant growth of the assembly happens before the enzymatic reaction dominates. Hence with higher equiv. of ATP (6.4 equiv.) and low concentrations of the enzyme (0.2–0.6 U), we could achieve the transient self-assembly of **1** as evident from the CD and scattering studies, evidently due to the difference in elongation and hydrolysis kinetics (Fig. 5b, Supplementary Fig. 36). With a higher unit of apyrase, there is faster hydrolysis of ATP, hence giving rise to faster disassembly as seen from faster decay in CD signal. In presence of 0.6 units of apyrase, the extent of elongation is also less as compared to 0.2 units of apyrase as seen from the frequency and amplitudes calculated from the CD spectra. This is observed due to the presence of higher units of enzyme, the hydrolysis supersedes elongation faster than for a lower unit of enzyme (Supplementary Table 6).

## Discussion

In conclusion, we have successfully shown a biomimetic strategy, which demonstrates time-dependent nucleation-growth and seeded supramolecular polymerization on selective interaction with a biologically relevant chiral auxiliary, ATP. This hypothesis has been proven experimentally as well as through detailed MM/MD simulations. The difference in lag phases obtained from various spectroscopic and scattering techniques gave us considerable insights into the processes involved in the formation of stable nuclei that elongates further. We also coupled fuel-driven aggregation with an enzymatic de-aggregation mechanism leading to growth under unstable regimes, which brings us closer to biological systems exhibiting dynamic instability. On a broader perspective, the results discussed here substantiate the importance of developing synthetic analogues to complex biological entities. On a course of future advancements, this would undoubtedly assist in rationalizing and thus incorporating temporally controlled materials and functions, respectively, into advanced materials. Furthermore, a single system showing seeded as well as transient characteristics via fuel-driven strategy exemplifies its highly modular nature. Therefore, a fuel-driven controlled supramolecular polymerization such as the present case could pave the path for the demonstration of numerous interesting properties in artificial systems, hence bringing us closer conceptually toward biological systems and phenomena.

## Methods

**General**. Detailed synthesis and characterization of **1** and the intermediates have been shown as Supplementary Methods in Synthesis and Characterization.

**Sample preparation**. The stock solution of molecule **1** was prepared in $CH_3CN$ ($5 \times 10^{-3}$ M) and required amount of stock and respective phosphates (Stock solution = $10^{-2}$ M) were injected into a 10 mM solution of HEPES buffer to make a total volume of 2.5 mL and achieve the desired concentration of the solution to be measured and the time-dependent elongation was monitored. Desired temperature was maintained throughout the measurements. For seeding experiments, the required amount of molecule **1** and ATP were injected into a solution of pre-grown stacks and the time-dependent elongation was monitored. For transient assembly measurements, to HEPES solution of apyrase, required amount of stock and ATP was added to it and the time-dependent elongation followed by disassembly was monitored. All the measurements were started immediately after all the required components are added.

**Computational details**. Calculation of atomic site charges: To carry out the MM/MD simulations of oligomers of either ATP-**1** or GTP-**1** in solution, the atomic site charges of either system have to be determined. The procedure to obtain the same is described below.

Firstly, a dimer configuration (a total of 291 atoms) was constructed using GaussView software[55] as shown in Supplementary Fig. 34. Density functional theory (DFT) calculations were performed using the QUICKSTEP module in CP2K software[56]. All valence electrons were treated in a mixed basis set with energy cutoff of 280 Ry. The short-range version of the double zeta single polarization basis set was used. The effect of core electrons was taken through pseudopotentials of Goedecker–Tetter–Hutter (GTH)[57]. The Perdew–Burke–Ernzerhof (PBE) exchange and correlation functional[58] was employed. DFT-D3[59] corrections were used to take van der Waals interactions into account and the dimer geometry was optimized in gas phase. The initial and final configurations of this gas phase dimer are shown in Supplementary Fig. 37, respectively.

A dimer of ATP-**1** has three parts: the OPV region, the DPA region, and the ATP. The phosphates of one ATP molecule can bind to two zinc atoms of two different molecules of **1**. A model of the oligomer wherein each ATP binds two adjacent molecules of **1** in the stacking direction would satisfy the two critical experimental observations: (a) the ends of the supramolecular polymer has free receptor sites to drive a "living" growth in presence of additional ATP and (b) the ATP to **1** mole ratio between 0.9 and unity. A schematic of such a model used for the MM/MD simulations is illustrated in Supplementary Fig. 38. The molecular structure of such a dimer is shown in Supplementary Fig. 41. Out of the four DPA receptor moieties in ATP-**1** dimer, two are bound to one ATP molecule, while the other two are free. Thus, the atomic site charges on either ends of ATP-**1** dimer will differ. Thus ATP-**1** (or GTP-**1**) oligomer will have both ATP-bound DPA ends as well as free (unbound) DPA receptor ends. Hence the force field has to have different charges on the backbone of **1** to take into account these structural differences. Supplementary Fig. 39a displays these different segments in ATP-**1** dimer, using which higher oligomers can be constructed. The atomic site charges for each segment in the dimer was determined through the DDEC/c3 method[60,61] using the electronic densities obtained via DFT calculation of the dimer, as described above. These segments were then fused to obtain site charges for higher oligomers. The procedure for such a construction is illustrated for the cases of trimer and hexamer in Supplementary Fig. 39b and c, respectively. The atom mapping scheme is provided in Supplementary Fig. 40 and the corresponding values of the charges on atoms of dimer are provided in Supplementary Tables 7–12.

**MM/MD simulations**. MD simulations were performed in all-atom representation. Molecule **1**, ATP, GTP, and counter ions were modelled using the DREIDING force field[62]. Water, used as solvent was modelled using the TIP3P force field[63]. Cross-interactions between the solute and solvent were considered through DREIDING mixing rules. A pseudo bond between Zinc and sp$^3$-hybridized nitrogen atom of DPA was created, with equilibrium bond length 2.2 Å, which was chosen from studying crystal structure of similar compounds[64]. Necessary number of counterions (Cl−) were added to the solution containing oligomers to attain charge neutrality. ATP-**1** oligomers containing the following number of OPV moieties were simulated in solution: 2, 3, 4, 6, 10, 15, and 25. For the case of GTP-**1**, only the 25-mer was studied. The initial configuration of the 25-mer of both ATP-**1** and GTP-**1** are shown in Supplementary Fig. 43. Details of the system sizes employed in MM/MD simulations of all oligomers are provided in Supplementary Table 13.

MD simulations were performed using LAMMPS package[65] at 298.15 K in the constant temperature and constant pressure ensemble. The Nosé–Hoover chain thermostat was used to maintain constant temperature and Nosé–Hoover barostat used to maintain constant pressure with a coupling constant of 1 ps. Three-dimensional periodic boundary conditions were employed. Non-bonded interactions were truncated at distance of 12 Å. Particle–particle particle–mesh (PPPM) solver was used to consider the long-range interactions. The equations of motion were integrated using the velocity Verlet integrator with a timestep of 0.5 fs. The coordinates of the molecules were stored for post-processing every 2.5 ps and

the trajectory was visualized using VMD[66]. In every simulation, the preformed oligomer was solvated in water using Packmol[67].

All MM/MD simulations of all the small oligomers were carried out for duration of 30 and 60 ns for 25-mer in GTP-**1** or ATP-**1** system. While the first 5 ns of the trajectory was used for equilibration, the structural analyses reported here were obtained from the last 25 ns of the trajectory for small oligomer size and last 55 ns for 25-mer.

**Data availability**. All data supporting the findings are available in the article as well as the supplementary information files and from the authors on reasonable request.

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

## Acknowledgements

We thank Professor C.N.R. Rao for kind support and encouragement. A.M. thanks JNCASR and DST for scholarship. D.B.K. thanks CSIR for fellowship. S.J.G. gratefully acknowledges SwarnaJayanti Fellowship Award (DST/SJF/CSA-01/2016-2017). A.M., S.J. G., and S.B. thank JNCASR and DST, Government of India and Sheikh Saqr Laboratory for financial support. We thank Dr. Reji Verghese (IISER Trivandrum) for AFM measurements.

## Author contributions

A.M. and M.K. performed the experimental work. A.M., M.K., A.J., and S.J.G. designed the concept and N.J. helped in the synthesis of the molecule **1**. A.M., A.J., and S.J.G. wrote the experimental part of the manuscript. D.B.K., K.K.B., and S.B. performed and analysed the MM/MD simulations, and wrote the theoretical part of the manuscript. All the authors discussed over the results and commented on the manuscript.

## Additional information

**Competing interests:** The authors declare no competing interests.

