## [Peer Review File · Nature Communications]

Reviewers' comments:

Reviewer #1 (Remarks to the Author):

Biomolecular active matter (driven by ATP or GTP hydrolysis) is fundamental to biology and to life itself, and hence the ability to synthesize biomimetic systems that exhibit such behavior is very important. The system described in this paper is quite interesting and the results are also important from the experimental point of view. However, I would suggest the following minor revisions:

(1) On p 2 at the beginning of the first full paragraph, the authors should mention that the assembly of G-actin into filaments also accelerates the rate of ATP hydrolysis by a factor of 40,000 due to changed molecular interactions and then cite this paper: M. McCullagh, M. G. Saunders, and G. A. Voth, "Unraveling the Mystery of ATP Hydrolysis in Actin Filaments" *J. Am. Chem. Soc.* 136, 13053–13058 (2014).

The authors also do some atomistic simulations. These are very short, and they see large scale structural rearrangements as far as I can tell, which is surprising but possible. There are 2 parts related to these simulations that should be checked and confirmed:

(2) The authors say that they add salt to neutralize the system. However, in general we add salts (KCl or NaCl) to mimic the salt concentration of the real system. However, I cannot tell what the salt concentration of the buffers in the experiment are. If it's not neutral, then the effect of changing this in solution should at least be checked.

(3) The authors define an inverse order parameter which is $\text{RMSD}/n\text{-oligomers}/\pi\text{-}\pi$ distance to quantify 'induced chirality'. First of all, this should be defined in the main text, not just the SI, since it is in a figure. Second, it's not clear to me why this is supposed to show induced chirality, and this should be further explained. It also goes to zero with $n\text{-oligomers}$, and it is not shown whether this is because RMSD decreases, or because $n\text{-oligomers}$ increases, or both. The authors should add this explicitly to the SI and discuss why they choose this order parameter, and what a low or high value really signifies.

In general, with the above revisions I find this interesting and important paper suitable for publication.

Reviewer #2 (Remarks to the Author):

The paper "Actin-mimetic temporal self-assembly via fuel-driven, controlled supramolecular polymerization" by the George group is a very interesting and novel piece of work. The present a dumbbell shaped molecule 1 consisting of an acceptor-donor-acceptor moiety flanked by two zinc complexes. The latter can chelate to the phosphate groups of adenosine triphosphate ATP. Molecule 1 in solution resides in a dormant state that does not significantly assemble. Upon binding of ATP, the dormant state is activated, and first achiral fibrous assemblies are formed, which shortly thereafter obtain significant supramolecular chirality. Once in the active state (1+ATP) a nucleation and elongation process occurs, resulting in a lag phase and a growth phase, respectively. Once assemblies are formed they can be used as seeds to further elongate the fibers upon addition of more 1+ATP (note: as a nice negative control, only addition of 1 in absence of ATP does not lead to elongation), where this time there is no lag phase. Control over the final size and dispersity is achieved by changing the ratio of initial seeds with respect to newly added 1+ATP. In a last step, the authors use the apyrase enzyme to hydrolyze ATP to ADP and AMP, the latter two of which are not able to maintain the active state, and molecule 1 is converted back into the dormant state. By using a large excess of ATP the authors can achieve transient self-assembly in line with their recent work on transient chirality.

First of all, I have to say that this is a very beautiful system with a rich behavior. The CD data, UV, Fluorescence, and light scattering are used to probe different aspects of the system, which allows a detailed view on what is happening. I do think the authors stress too much the similarities with actin polymerization. The only similarities are that ATP is non-covalently binding to G-actin monomer (i.e., similar to 1+ATP), and that a nucleated self-assembly is present. F-actin (fibers), however, have different growth rates at the + vs. the - end, the actin itself is the enzyme that hydrolyzes ATP, and conformational changes are occurring. So, is this that similar to stress throughout the entire manuscript that it's just like actin? I do not think this is necessary and this could be toned down. The work in itself is nice enough to be self-standing. Cooperative self-assembly in a fuel-driven system has been demonstrated by Leira-Iglesias et al (ChemComm, 2016, 52, 9009-9012), and should be cited. The latter is sufficiently different and does not impact the novelty of the current work, which merits publication in NCOMM.

There are, however, a number of specific technical points that need attention before proceeding with publication.

1) The authors use two different models to explain the cooperative polymerization of their system. In the first part of the manuscript the Watzky and Finke model, and later on the more familiar Oosawa model. Both are nucleation/elongation models, and I do not see why they are both used. In Ref. 1 of the manuscript (Morris, Watzky, Finke, 2009) there is a nice overview of available models, so why not use one of the two. In addition, the Watzky/Finke model is a quite vague phenomenological model, which can describe nearly any sigmoidal time curve. Given the level of detail at which the system has been studied it would suit the authors to develop a simple model (e.g., expansion of the Oosawa model) that better captures the behavior, and allows for real insights into the molecular level.

2) The sentence: "Increase in the values of t_{lag} and t_{50} (half time i.e. time required for completion of 50% of the process)". I think the authors mean "decrease" instead of "increase".

3) A minor point. Fig. 2e,f: it would be good to explain a bit more what can be seen and what can be concluded from this. I find it hard to follow.

4) Fig. 3h: why are the distributions of π - π distance bimodal?

5) Fig. 4a: how much material (1+ATP) is added at each time point (please specify in the caption)?

6) Many values that were obtained from fitting do not have a standard deviation (e.g., supplementary table 1, 2, 4, 5, etc...). Without that it is difficult to judge the validity of the claim that the lag times of the polymerization as measured by the different experimental techniques are significantly different. The same goes for error bars in figures like Fig. 4e, Supp. Fig. 4, 9, 13, 28.

7) The main issue I have is with the experiments done to prove that length-control and narrow polydispersity can be obtained.

a. To this end, many light scattering experiments were performed using a Malvern NanoZS instrument with a fixed (back scattering) angle of 173 degrees. The main text and supplementary info states that dynamic light scattering (DLS) is used. To my knowledge (but I am not an expert on this Malvern instrument), the machine obtains a correlation function, giving a diffusion time, which is converted to sizes using a spherical model. That is, the machine always assumes that spherical particles are present. While this is fine for simple colloids or vesicles, the authors are working with fibers of high aspect ratio, so far from spherical in my opinion. The trends in size when changing $[1]/[S]$ should still be valid, but the numbers would (if my assumptions are right) be wrong.

b. The polydispersity is also obtained from the NanoZS instrument. It is impossible to obtain the number average chain length L_n and weight average chain length L_w from DLS. I am assuming the authors used static light scattering SLS, but for that to be done properly dn/dC (where n is the refractive index and C the concentration) has to be known, and to obtain reliable data the scattering intensity should be collected at multiple angles, which does not seem the case here. In addition, supplementary figure 28 (the inset showing PDI vs. $[1]/[S]$) does not have error bars, so it is difficult to substantiate the claim of narrow polydispersity in my opinion. Some measurements on a multi-angle setup (i.e., with goniometer or multiple detectors), like the ones made by ALV, LS-Instruments, or others would be needed to benchmark these results. Or if the authors can use

another technique to prove the PDI values (perhaps GPC if the assemblies are stable enough)?
8) In the supplementary information, when aliquots are added (e.g., Fig. 19) the caption should state the volume and concentration.

In my printed version of Fig. 1, a square boxes appear in the background, which are likely due to file conversion. Finally, as a minor detail; quotation marks should never be used to emphasize certain concepts like e.g. just before the Results and Discussion section: a general strategy towards "controlled" supramolecular polymerization.... This could be interpreted as it being not controlled at all. Other occurrences in the manuscript should be removed as well.

Overall, this is exciting work and I highly recommend publication in NCOMM.

Reviewer #3 (Remarks to the Author):

This manuscript by Mishra et al. presents results on a synthetic, self-assembling system. The authors make multiple comparisons to biological systems, most notably actin, since the assembly is ATP-dependent and the system exhibits a nucleation-elongation behavior similar to actin assembly. Although this comparison is apt in some respect, it falls short in many others, and these parts of the work raise questions.

1. The assembly data presented is not direct observation of nucleation or polymerization. The actin field has used pyrene fluorescence for decades since this has been shown to directly correspond to assembly, but the CD, absorbance, DLS data, etc. that is presented has not been shown to have the same correspondence. Use of a fluorescent ATP analog might allow for direct observation of nucleation and growth and that would be a significant addition to this work.

2. To make further comparisons with actin or to simply have a meaningful kinetic model for this process, the authors really need to look at the concentration dependence of this process. This could be done using seeds (as on p. 10) to determine the kinetic growth rate parameter. Figure 4e appears to be along these lines, but the details are not presented. Determining this would then allow the nucleation process to be further teased out once this one part of the model was fixed. It would also be useful to know if there was a critical nucleus size, and more detailed studies would also allow that to be determined.

3. Perhaps one of the biggest issues I see relates to the how hydrolysis affects polymerization. Unlike biological systems, it appears that the nucleotide is **not** hydrolyzed naturally within the polymer that is formed, but rather this is only accomplished using external enzyme. This might suggest that there is some level of nucleotide exchange happening and/or the enzyme is acting directly on the polymer to cause disassembly? Having a synthetic system that utilized hydrolysis would be very unique, but this system does not have that feature. In this sense, this system is not that different from any other polymerizing system that is induced by ions or other stimuli.

Response to Reviewer Comments: NCOMMS-17-22768-T

(A. Mishra, S. J. George, et al.)

Reviewer #1 (Remarks to the Author):

Reviewer Remarks: *Biomolecular active matter (driven by ATP or GTP hydrolysis) is fundamental to biology and to life itself, and hence the ability to synthesize biomimetic systems that exhibit such behaviour is very important. The system described in this paper is quite interesting and the results are also important from the experimental point of view. However, I would suggest the following minor revisions:*

Response:

We thank the reviewer for the positive comments. The data presented here, as pointed out by the referee, is unique as ability to synthesize biomimetic systems that exhibit fuel-driven temporally controlled supramolecular polymerization behaviour is indeed scarce.

Reviewer Comment 1: *On p 2 at the beginning of the first full paragraph, the authors should mention that the assembly of G-actin into filaments also accelerates the rate of ATP hydrolysis by a factor of 40,000 due to changed molecular interactions and then cite this paper: M. McCullagh, M. G. Saunders, and G. A. Voth, "Unravelling the Mystery of ATP Hydrolysis in Actin Filaments" J. Am. Chem. Soc. 136, 13053–13058 (2014). The authors also do some atomistic simulations. These are very short, and they see large scale structural rearrangements as far as I can tell, which is surprising but possible. There are 2 parts related to these simulations that should be checked and confirmed:*

Response: A modified version of the sentence mentioned by the reviewer has been included in the revised version of the manuscript and the respective citation has been done (as reference number 5).

Regarding the concerns of duration of the simulations, the MD simulations have now been extended up to 30 ns for all the systems. In the case of the 25-mer, it has been extended to 60 ns. Results reported in the revised manuscript have been averaged over the 5 to 30 ns window of the trajectory for the shorter oligomers and over the 5 to 60 ns window of the trajectory for the 25-mers of ATP-1 and GTP-1. The revised results closely match what was reported in the original manuscript.

We have calculated several parameters like, twist angle between the consecutive molecules in a stack, π - π distance and RMSD values. The convergence of these parameters allows us to conclude our results from the 30 ns trajectory. The atomistic simulations have been carried out to understand all the processes involved in the rearrangement of molecule **1** which facilitates the formation of an active nuclei which further elongates.

Reviewer Comment 2: *The authors say that they add salt to neutralize the system. However, in general we add salts (KCl or NaCl) to mimic the salt concentration of the real system. However, I cannot tell what the salt concentrations of the buffers in the experiment are. If it's not neutral, then the effect of changing this in solution should at least be checked.*

Response: Experimentally, the system is charge neutral. We added counter ions in the MD simulations only to have a charge-neutral system, and these ions are synonymous to the ions present under experimental conditions. To reiterate, excess salt has not been added. Furthermore, in experiments, ZnClO_4 has been used for the complexation of the molecule, thus making ClO_4^- as the counter ion. However in our simulations, a simpler and smaller Cl^- has been used instead of ClO_4^- as the counter ion. In order to confirm that the counter ions do not influence the fuel-driven seeded assembly characteristics of **1**, we have now synthesized molecule **2** with Cl^- counter ions and have studied its self-assembly properties. This molecule too showed similar self-assembly behaviour as that of **1**, which suggests that the counter ion does not affect the nucleation-elongation property as well as the seeded self-assembly, hence justifying the use of Cl^- ions in the MD simulations. This has been added to the revised version as Supplementary Figure 22.

Supplementary Figure 22] **a**, Molecular structure of **2**. **b**, CD and **c**, UV-Vis absorbance spectra represent time lapse spectra of ATP-driven growth of **2**. **d**, CD signal showing seeded supramolecular polymerization of **1** upon a subsequent addition of monomer along with ATP ($c = 2 \times 10^{-5}$ M, HEPES/CH₃CN, 90/10, v/v, 0.9 equiv. ATP, 30 °C).

Reviewer Comment 3: *The authors define an inverse order parameter which is RMSD/ n -oligomers/ π - π distance to quantify 'induced chirality'. First of all, this should be defined in the main text, not just the SI, since it is in a figure. Second, it's not clear to me why this is supposed to show induced chirality, and this should be further explained. It also goes to zero with n -oligomers, and it is not shown whether this is because RMSD decreases, or because n -oligomers increases, or both. The authors should add this explicitly to the SI and discuss why they choose this order parameter, and what a low or high value really signifies.*

Response: The Root Mean Square Deviation (RMSD) exhibits a weak decrease with increase in the oligomer size which in turn leads to much lower values of the Inverse Order Parameter (IOP) over oligomer size. We have performed simulations on systems of various oligomer sizes, and the initial geometry of all the oligomers were constructed with the same twist angle (i.e. 25°), which indeed induced helicity in the stack. During the MD simulations, the molecules in shorter oligomers exhibit structural differences with this initial configuration, such as slip between the consecutive molecules, as discussed in the Supplementary Table 14 and a deviation in the twist angle. The RMSD provides a measure of the fluxionality of the oligomer. It has been calculated with respect to a well equilibrated configuration, chosen arbitrarily (in fact three such reference structures from the trajectory were chosen so as to demonstrate that the RMSD is independent of this choice. See figure below) The RMSD has been added as a new figure in the revised Supplementary Information (Supplementary Figure 17). The RMSD exhibits a weak decrease with increasing oligomer size. We expect RMSD itself to decay slowly with oligomer size, for stack sizes longer than those studied here. RMSD is a 'distance'. In order to obtain a dimensionless quantity, the same was divided by the mean π - π distance of the 25-mer (i.e., 3.8 Å) and the number consecutive pairs in an assembly, which is $(n-1)$. The IOP per se does not imply induced chirality. Lower values of IOP imply lower fluxionality and greater stability of the stack.

Keeping in line with the concerns raised by reviewer 2 also on this regard, a more detailed discussion has been added in the main text explaining Figure 2 e and f.

Supplementary Figure 17 | RMSD of various oligomers extracted from MM/MD simulations of ATP-1 stacks with reference frame chosen as follows: **a**, at 28 ns and **b**, at 29 ns and **c**, at 30 ns. IOP has been calculated from the RMSD values using the following expression. $IOP = RMSD / ((n-1) * \pi - \pi \text{ distance})$

Reviewer #2 (Remarks to the Author):

Reviewer Remarks: *The paper “Actin-mimetic temporal self-assembly via fuel-driven, controlled supramolecular polymerization” by the George group is a very interesting and novel piece of work. The present a dumbbell shaped molecule 1 consisting of an acceptor-donor-acceptor moiety flanked by two zinc complexes. The latter can chelate to the phosphate groups of adenosine triphosphate ATP. Molecule 1 in solution resides in a dormant state that does not significantly assembly. Upon binding of ATP, the dormant state is activated, and first achiral fibrous assemblies are formed, which shortly thereafter obtain significant supramolecular chirality. Once in the active state (1+ATP) a nucleation and elongation process occurs, resulting in a lag phase and a growth phase, respectively. Once assemblies are formed they can be used as seeds to further elongate the fibers upon addition of more 1+ATP (note: as a nice negative control, only addition of 1 in absence of ATP does not lead to elongation), where this time there is no lag phase. Control over the final size and dispersity is achieved by changing the ratio of initial seeds with respect to newly added 1+ATP. In a last step, the authors use the apyrase enzyme to hydrolyze ATP to ADP and AMP, the latter two of which are not able to maintain the active state, and molecule 1 is converted back into the dormant state. By using a large excess of ATP the authors can achieve transient self-assembly in line with their recent work on transient chirality.*

First of all, I have to say that this is a very beautiful system with a rich behavior. The CD data, UV, Fluorescence, and light scattering are used to probe different aspects of the system, which allows a detailed view on what is happening. I do think the authors stress too much the similarities with actin polymerization. The only similarities are that ATP is non-covalently binding to G-actin monomer (i.e., similar to 1+ATP), and that a nucleated self-assembly is present. F-actin (fibers), however, have different growth rates at the + vs. the –

end, the actin itself is the enzyme that hydrolyzes ATP, and conformational changes are occurring. So, is this that similar to stress throughout the entire manuscript that it's just like actin? I do not think this is necessary and this could be toned down. The work in itself is nice enough to be self-standing. Cooperative self-assembly in a fuel-driven system has been demonstrated by Leira-Iglesias et al (ChemComm, 2016, 52, 9009-9012), and should be cited. The latter is sufficiently different and does not impact the novelty of the current work, which merits publication in NCOMM.

Response: We thank the reviewer for very positive and constructive comments and we agree with the concerns regarding the similarities that we claim our system has with actin. Though there are short comings, this is indeed the first time a system so close to temporal aspects of actin self-assembly has been reached in a synthetic molecule. However we agree with the reviewer on the actin mimetic nature and hence we have toned down our claims for “actin mimetic self-assembly” throughout the manuscript. We also agree to the reviewer that the novelty of this piece of work would be the concept of fuel-driven cooperative self-assembly similar to that of biological systems and we have also cited the *Chem Comm* **52**, 9009-9012 (2016) (as reference number 40) which is indeed an important piece of work in this direction. We have also removed the “Actin-Mimetic” term from the title of the manuscript in this revised version.

Reviewer Comment 1: *The authors use two different models to explain the cooperative polymerization of their system. In the first part of the manuscript the Watzky and Finke model, and later on the more familiar Oosawa model. Both are nucleation/elongation models, and I do not see why they are both used. In Ref. 1 of the manuscript (Morris, Watzky, Finke, 2009) there is a nice overview of available models, so why not use one of the two. In addition, the Watzky/Finke model is a quite vague phenomenological model, which can describe nearly any sigmoidal time curve. Given the level of detail at which the system has been studied it would suit the authors to develop a simple model (e.g., expansion of the Oosawa model) that better captures the behaviour, and allows for real insights into the molecular level.*

Response: We thank the reviewer for the comment. We also agree to reviewer about the importance of developing a model for fuel-driven synthetic systems to simultaneously analyse the cooperative assembly and seeded self-assembly. However, at this point we have the lack of expertise for doing the same and would require a collaborative effort. We feel this is beyond the scope of the present manuscript and will definitely address this issue in the near future. In the current work we have used two different equations for fitting two

different processes (i. e. Watzky-Finke equation for nucleation-elongation and equation proposed by Zhao and Moore for seeded assembly) which have been utilised previously because we need to derive different parameters from both the equation for two different processes. Hence two general equations for probing two separate processes have been used in the manuscript. A more detailed justification is given below.

The Watzky-Finke equation has been used to fit the nucleation-elongation curve for various proteins like amyloid and prion proteins where they follow a sigmoidal growth corresponding to cooperative elongation. The parameters derived from this equation are k_n (rate of nucleation) and k_e (rate of elongation).

While for fitting the seeded assembly we have used a modified equation derived from the kinetic model developed for actin polymerization.

The initial kinetic model for actin polymerization was developed by Oosawa and Kasai (*Biochim. Biophys. Acta* 57, 22-31 (1962)) where the differential equation used by them was:

$$\frac{dc_t^*}{dt} = -\frac{d[M]}{dt} = (k[M] - k_-)c_p^* + n\frac{dc_p^*}{dt} + k_-[M_n]$$

where c_p^* is the total concentration of all polymers equal to or larger than nucleus size n and c_t^* is the concentration of monomers incorporated into these polymers. (k^* and k_-^* are rate constants for formation and dissociation of nuclei).

Seeded polymerization occurs when the rate of nucleation is exceptionally slow and elongation is very fast ($k^* \rightarrow 0$ and $k \gg k_-$). Hence the previous equation reduces to:

$$\frac{dc_t^*}{dt} = -\frac{d[M]}{dt} = (k[M] - k_-)c_p^*$$

where c_p^* is a constant determined by the concentration of seeds added. Integration of the above equation gives:

$$[M] - [M]_\infty = ([M]_0 - [M]_\infty) \exp(-k_{e(\text{seeded})}[S]t)$$

which is the equation used by us. $[M]_0$ is the concentration of fresh feed added to a particular amount of seed $[S]$. $[M]_\infty$ corresponds to concentration of monomers coexisting with supramolecular polymer at equilibrium. $[M]$ corresponds to the concentration of monomers at any time t elongating with a rate constant of $k_{e(\text{seeded})}$. The parameter derived from this equation is $k_{e(\text{seeded})}$. k_n does not exist in a seeded assembly. This modified equation was proposed by Zhao and Moore in (*Org. Biomol. Chem.* 1, 3471-3491 (2003)) which has also been used by Würthner and co-workers for fitting their seeded assembly data in *J. Am. Chem. Soc.* 137, 3300-3307 (2015). Though the origin of this equation is from equations derived for self-assembly of actin, this is a generalized equation used for seeded assemblies.

Reviewer Comment 2: *The sentence: “Increase in the values of t_{lag} and t_{50} (half time i.e. time required for completion of 50% of the process”. I think the authors mean “decrease” instead of “increase”.*

Response: We thank the reviewer for pointing out this mistake and corrected in the revised manuscript.

Reviewer Comment 3: *A minor point. Fig. 2e,f: it would be good to explain a bit more what can be seen and what can be concluded from this. I find it hard to follow.*

Response: Fig 2e, f have been explained in more details in the main text as suggested by the Reviewer and which also takes care of the comments of Reviewer 1 in this regard.

Reviewer Comment 4: *Fig. 3h: why are the distributions of π - π distance bimodal?*

Response: The reason for the bimodal distribution of π - π distance is as follows: The peripheral molecules in the stack are more fluxional having larger π - π distances, than the ones in the core of the stack. We expect the intensity of the feature at larger distance to decrease with increasing stack size. This behaviour produces the bimodality. In addition to this, in case of GTP-1 system, the molecules in stack are not packed properly. Due to this, we see a multimodal distribution in GTP-1 system. This is now explained in the figure caption for Figure 3h.

Reviewer Comment 5: *Fig. 4a: how much material (1+ATP) is added at each time point (please specify in the caption)?*

Response: In every figure where ATP is added, the concentration of 1 and ATP has been mentioned in the figure caption of the revised text

Reviewer Comment 6: *Many values that were obtained from fitting do not have a standard deviation (e.g., supplementary table 1, 2, 4, 5, etc...). Without that it is difficult to judge the validity of the claim that the lag times of the polymerization as measured by the different experimental techniques are significantly different. The same goes for error bars in figures like Fig. 4e, Supp. Fig. 4, 9, 13, 28.*

Response: We thank the reviewer for this valid suggestion and wherever data have been got by fitting; the standard deviations are now mentioned in the tables of this revised version.

It can be seen that the claims made regarding lag phases still holds even after adding standard deviation values. Similarly error bars have also been added into the data wherever applicable.

Reviewer Comment 7: *The main issue I have is with the experiments done to prove that length-control and narrow polydispersity can be obtained.*

a. To this end, many light scattering experiments were performed using a Malvern NanoZS instrument with a fixed (back scattering) angle of 173 degrees. The main text and supplementary info states that dynamic light scattering (DLS) is used. To my knowledge (but I am not an expert on this Malvern instrument), the machine obtains a correlation function, giving a diffusion time, which is converted to sizes using a spherical model. That is, the machine always assumes that spherical particles are present. While this is fine for simple colloids or vesicles, the authors are working with fibers of high aspect ratio, so far from spherical in my opinion. The trends in size when changing $[1]/[S]$ should still be valid, but the numbers would (if my assumptions are right) be wrong.

Response: We agree with the reviewer that DLS used, gives data modelled on spherical particles and for unsymmetrical aggregates this data corresponds to its hydrodynamic diameter. We would argue that fibres generated in solution are actually not of high aspect ratio as evidence from smaller fibres from TEM. Thus we believe that fibres in solution are still small enough to fall under a rough assessment of length by DLS. We agree however that this is still not a very exact measurement and is intended to give a trend of size increase in a controlled manner. So instead of trying to comment on the exact values of sizes we would like to comment on the trends which match our expectations. Hence we have removed the calculations of L_n from DLS from the main text and has been replaced with the increase of hydrodynamic diameter versus $[1]/[S]$, (inset of Figure 4f), showing a linear fit of the size increase.

Figure 4f] DLS data showing the increase in hydrodynamic radii from seed (S) to varying $[1]/[S]$ ratios (Inset shows the straight line fit of size increase for various seed concentrations where green ball is the seed size and blue balls are the sizes of seeded stacks) ($[S] = 1 \times 10^{-5}$ M, $[1] = 0.33 \times 10^{-5}$ M (1.65 μ L), 0.5×10^{-5} M (2.5 μ L), 1×10^{-5} M (5 μ L), 2×10^{-5} M (10 μ L), 5.7×10^{-5} M (2.85 μ L) (HEPES/CH₃CN, 90/10, v/v, 0.9 equiv. ATP, 30 °C).

b. The polydispersity is also obtained from the NanoZS instrument. It is impossible to obtain the number average chain length L_n and weight average chain length L_w from DLS. I am assuming the authors used static light scattering SLS, but for that to be done properly dn/dC (where n is the refractive index and C the concentration) has to be known, and to obtain reliable data the scattering intensity should be collected at multiple angles, which does not seem the case here. In addition, supplementary figure 28 (the inset showing PDI vs. $[1]/[S]$) does not have error bars, so it is difficult to substantiate the claim of narrow polydispersity in my opinion. Some measurements on a multi-angle setup (i.e., with goniometer or multiple detectors), like the ones made by ALV, LS-Instruments, or others would be needed to benchmark these results. Or if the authors can use another technique to prove the PDI values (perhaps GPC if the assemblies are stable enough)?

Response: We completely respect reviewer's concern on calculation of PDI from DLS experiment. Hence we have removed the calculations of L_n and L_w from DLS experiments from the main text of the manuscript. Though there might be errors in the absolute values due to system constraints the conclusions made in the work are based on emerging trends. So instead we have plotted the size changes versus the fresh feed concentration as mentioned in the previous answer. As mentioned in the previous comment the fibres generated in solution are actually not of high aspect ratio as evidence from smaller fibres from TEM. Thus we believe that fibres in solution are still small enough to fall under a rough assessment of size by DLS.

We have also included the dispersity trends obtained directly from DLS (without any length statistics) which also shows a controlled supramolecular polymerization reaction. We would also like to add that Correlation curves from DLS show that the data recorded is of “good quality” confirming the fact the fibres produced in solution are perhaps much smaller and do not deviate from the fitting model by great extent.

To answer the reviewer’s question in a better way we tried the SLS experiments, but unfortunately due to a combination of factors (long time molecular stability of sample in solution as the time taken for multi angle SLS is more, lack of strong scattering) these did not fructify. At this point we would also like to reiterate that length control is not a feature in biological systems like actin, however since we saw indications of length control from our data, considering this to be an additional and curiously interesting property, we have reported it in our manuscript. The lack of consolidated and direct evidence (like cryo-TEM, SLS) by no means should diminish the fuel driven seeded characteristic of our system.

So we have modified the Supporting figures in the following manner.

Supplementary Figure 30] Parameters derived from DLS of seeding experiments. a, Dispersity value derived directly from DLS with an average value calculated as 0.07 showing a narrow polydispersity of the system. **b,** Correlation function change for unseeded (black curve) and seeded (green for $[1]/[S] = 0.33$ and orange for $[1]/[S] = 2$) data shows an increase in lag time on seeding showing an increase in size with increase in scattering on seeding.

We have also kept the calculation of PDI values from the DLS distributions as a separate figure in ESI to show the trend and we have clearly mentioned that absolute values should not be considered from this data.

Supplementary Figure 31| Parameters derived from DLS of seeding experiments.

Weight-average length (L_w) (star, green = $[S]$, blue = $[S]+x[1]$), number-average length (L_n) (filled circles, pink = $[S]$, red = $[S]+x[1]$). Inset shows the PDI (L_w/L_n) as a function of the ratio of the total amount of added fresh feed to the initial amount of seed. ($[S] = 1 \times 10^{-5}$ M, $[1] = 1 \times 10^{-5}$ M, $\text{CH}_3\text{CN}/\text{HEPES}$, 10/90, v/v, 0.9 equiv. ATP, 30 °C). These data indicate that both L_w and L_n follow a similar linear trend in increase with increasing $[1]/[S]$ ratio. The polydispersity index (PDI) of the system also remains constant with an average value of 1.1 suggesting a good control over the degree of polymerization and dispersity. Though the L_w and L_n values obtained from DLS distribution is not conventional for anisotropic structures, we would like to specify that we are interested only in the trends obtained and not the absolute values and hence presented the data in Supplementary Fig. 31. In addition the fibres generated in solution are actually not of high aspect ratio as evidence from smaller fibres from TEM. Thus we believe that fibres in solution are still small enough to fall under a rough assessment of size by DLS. Although we have tried the SLS experiments, but unfortunately due to a combination of factors (long time molecular stability of sample in solution as the time taken for multi angle SLS is more, lack of strong scattering) these experiments did not fructify.

Reviewer Comment 8: *In the supplementary information, when aliquots are added (e.g., Fig. 19) the caption should state the volume and concentration.*

Response: Wherever figures mention the addition of aliquots into the solution, the volume and concentration of both molecule 1 and ATP has been mentioned in the captions.

Reviewer Comment 9: *In my printed version of Fig. 1, square boxes appear in the background, which are likely due to file conversion. Finally, as a minor detail; quotation marks should never be used to emphasize certain concepts like e.g. just before the Results and Discussion section: a general strategy towards “controlled” supramolecular polymerization.... This could be interpreted as it being not controlled at all. Other occurrences in the manuscript should be removed as well.*

Response: We thank the reviewer for his comment and we have taken care of the issues raised by them.

Reviewer #3 (Remarks to the Author):

Reviewer Remarks: *This manuscript by Mishra et al. presents results on a synthetic, self-assembling system. The authors make multiple comparisons to biological systems, most notably actin, since the assembly is ATP-dependent and the system exhibits a nucleation-elongation behavior similar to actin assembly. Although this comparison is apt in some respect, it falls short in many others, and these parts of the work raise questions.*

Response: Naturally occurring self-assembled systems are extremely complex and their assembly involves a series of interactions which give rise to interesting structures further giving rise to important functions. Scientists have recently started to design artificial systems that understand and employ the natural processes involved in biological self-assembly, which is the stepping stone to creating multi-functional materials with structural and temporal control. In this manuscript we describe a synthetic system which is closest to natural actin from the structural point of view. Though still far from coming close to actin in terms of the function, we have tried to achieve some similarities between the self-assembly process of actin through our small molecule, **1**. However, considering the comments from Reviewer 2 as well, we have toned down the comparison of our self-assembly with that of Actin, in this revised manuscript and also modified the title and abstract, accordingly.

Reviewer Comment 1: *The assembly data presented is not direct observation of nucleation or polymerization. The actin field has used pyrene fluorescence for decades since this has been shown to directly correspond to assembly, but the CD, absorbance, DLS data, etc. that is presented has not been shown to have the same correspondence. Use of a fluorescent ATP analog might allow for direct observation of nucleation and growth and that would be a significant addition to this work.*

Response: Actin requires labelling with various fluorescent chromophores because by itself it does not contain any group which can provide good spectroscopic signatures. This allows the monitoring of nucleation-elongation process of actin through fluorescence changes of fluorescent probes like pyrene which helps in monitoring the self-assembly process of actin. Now, our molecule consists of a chromophore which is conjugated and consists of an extensive π -surface, which provides a sensitive spectroscopic probe for the self-assembly due to inter-chromophoric interactions. Hence, we are directly able to monitor the self-

assembly through methods like absorbance, fluorescence and circular dichroism. Moreover, our system is a small molecule assembly and hence addition of pyrene (or any other labelling chromophore) would affect the self-assembly process due to competing weak interaction like π - π interactions and hence it is not possible to perform such experiments. However, as suggested by the reviewer we have tried the self-assembly of **1** using fluorescently labelled ATP in an attempt to visualize the assembly process. However, in the present case, the growth of molecule **1** is highly selective to ATP due to the optimum number of required intermolecular hydrogen bonds which stabilizes the stacks. As a result the labelled ATP (Mant-ATP) hinders the self-assembly of the molecule and does not show any elongation as shown in g. Hence, a fluorescent ATP could not be used to visualize the growth in the present case.

a, Chemical structure of Mant-ATP, **b**, CD intensity, **c**, absorbance spectra and **d**, CD spectra of **1** with ATP and Mant-ATP ($c = 2 \times 10^{-5}$ M, HEPES/CH₃CN, 90/10, v/v, 0.9 equiv. ATP and Mant-ATP, 30 °C).

Reviewer Comment 2: *To make further comparisons with actin or to simply have a meaningful kinetic model for this process, the authors really need to look at the concentration dependence of this process. This could be done using seeds (as on p. 10) to determine the kinetic growth rate parameter. Figure 4e appears to be along these lines, but the details are not presented. Determining this would then allow the nucleation process to be further teased out once this one part of the model was fixed. It would also be useful to know if there was a critical nucleus size, and more detailed studies would also allow that to be determined.*

Response to the comment:

In actin, aggregation occurs only above critical concentration of F-actin. The critical concentration defined in actin is the effective concentration of F-actin remaining due to the difference in rates of assembly and disassembly and the concentration reached once rate of assembly overcomes disassembly which can undergo elongation. Rate of elongation increases with increasing concentration of G-actin - shown by Oosawa (M. Kasai, S. Asakura and F. Oosawa, *Biochim. Biophys. Acta*, 57 (1962) 22-31). We have tried for two different concentrations i.e. 1×10^{-5} M and 2×10^{-5} M and we see that there is an increase in rate of elongation with increasing concentration. But unfortunately, beyond 2×10^{-5} M, the self-assembly shows anomalous behaviours because of solubility issues of higher concentration of the system which causes precipitation. The logarithm plot of apparent rate of growth calculated from the absorbance changes versus the concentration of added monomers gives a linear relation (Fig. 4e), as mentioned by the reviewer, indicates that the supramolecular polymerization reaction is of "first order" with respect to the fresh feed concentration which concludes a chain growth-like seeded supramolecular polymerization. The critical nucleus has been defined as "The smallest aggregate for which [the rate of] elongation is faster than [the rate of] dissociation" A. Wegner, J. Engel. Kinetics of the cooperative association of actin to actin filaments, *Biophys. Chem.* 3 (1975) 215–225. Unfortunately, for a synthetic system like ours, kinetic models have not been developed which can derive the kinetic growth rate parameters as well as the critical nucleus sizes as these are small molecule assemblies.

Reviewer Comment 3: *Perhaps one of the biggest issues I see relates to the how hydrolysis affects polymerization. Unlike biological systems, it appears that the nucleotide is *not* hydrolyzed naturally within the polymer that is formed, but rather this is only accomplished using external enzyme. This might suggest that there is some level of nucleotide exchange happening and/or the enzyme is acting directly on the polymer to cause disassembly? Having a synthetic system that utilized hydrolysis would be very unique, but this system does not have that feature. In this sense, this system is not that different from any other polymerizing system that is induced by ions or other stimuli.*

Response: We thank the reviewer for the comments and agree to the concerns regarding the fuel driven dis-assembly of the supramolecular polymers. It is true that we need to add an external catalyst which hydrolyses ATP to ADP causing its disassembly, unlike nature's self-catalysed process. However, we would like to point out that we have been able to achieve an actin mimetic fuel-driven ATP-selective assembly in a synthetic system for the first time, which can temporally dis-assemble on enzymatic hydrolysis of the ATP. And this is indeed different from the other stimuli-responsive materials which act in a passive manner,

whereas the present system operate under non-equilibrium to get an active assembly. This has been achieved by a clever temporal programming of the activator and deactivator fuels. Even in the presence of apyrase, the molecule first assembles as the rate of elongation is more than hydrolysis. But as soon as that rate overcomes the rate of hydrolysis, it starts disassembling. Hence, we are able to achieve an ATP-fuelled transient assembly of molecule **1** even though it is not self-catalysed. A self-hydrolysing stack is the next level of challenge and work on that aspect is under progress in our group using guanidinium based receptors.

To address the concerns of the reviewer as to whether the enzyme is directly attacking the ATP bound to the stack or there is some nucleotide exchange happening we would like to refer to our previous work wherein we have looked into this using a Chen's phosphate assay. When the kinetics of phosphate release on ATP hydrolysis monitored in the absence of molecule is much faster as compared to in the presence of molecule (Supplementary Figure 24 and 25, *Nat. Commun.* **5**, 5793 (2014)). Further experiments with excess of multiple nucleotides did not show any selective hydrolysis of weakly associating nucleotides in solution. These results suggest that the enzyme acts on the ATP bound on to the stacks though we does not rule out an exchange of nucleotides, as we have also shown a fast nucleotide exchange in this system using competitive gust binding experiments.

A similar study has been now performed in the present system in presence of excess of ADP (Supplementary Fig. 32b). To understand if enzyme selectively hydrolyses the unbound or bound phosphates, CD kinetic measurements of ATP-1 in presence of other phosphates (0.9 equiv. of ATP followed by addition of 0.9 equiv. of ADP) were performed. The decrease in CD signal with time is clear indication of the dis-assembly of the stacks because of the hydrolysis of ATP. As expected, due to competitive binding, ATP should be bound to molecule **1** to form ATP-1 stacks, whereas ADP must be free in solution. Gradual decrease in signal immediately after completion of elongation without a lag, suggests that apyrase has no preferential action to unbound phosphates compared to bound ones. If the enzyme were to preferentially act on unbound phosphates, a constant CD signal would have been obtained initially till all unbound phosphates are consumed and only then signal should have started to decrease. These results suggest that the enzyme acts on the ATP bound on to the stacks though we does not rule out an exchange of nucleotides, as we have already shown a fast nucleotide exchange on similar system in our previous studies, using competitive gust binding experiments.

This has been now mentioned in the main text and added the Figure in the ESI (Supplementary Fig. 32b).

Supplementary Figure 32 b| Time dependent variation in CD signal monitored at 500 nm of ATP-1 grown stacks in the presence of 0.9 equiv. of ADP monitored at 500 nm, on introducing 7.2 units of the enzyme potato apyrase. To understand if enzyme selectively hydrolyses the unbound or bound phosphates, CD kinetic measurements of ATP-1 in presence of other phosphates (0.9 equiv. of ATP followed by addition of 0.9 equiv. of ADP) were performed. ($[1] = 2 \times 10^{-5} \text{ M}$, $\text{CH}_3\text{CN}/\text{HEPES}$, 10/90, v/v, 30°C).

Brief note for all the reviewers:

From extended simulation trajectory, we updated Figure 2e, 2f, 3f, 3g and 3h and Supplementary Figs. 16 and 17 have been added with RMSD plots and IOP plots were changed. Along with that, reference nos. 5, 40, 44, 45, 55-68 have been added in the main text and reference no. S12 has been added in the Supplementary Info.

Figures 4e and 4f have been updated from the main text where error bars were added and the plot corresponding to \ln versus $[1]/[S]$ was replaced by size versus $[1]/[S]$. Supplementary Fig. 30 summarizes the parameters derived directly from DLS. Along with that, Supplementary Figs. 4, 9, 13, 28 and Supplementary Tables 1, 2, 3, 4 and 5 were updated with standard deviation values and error bars.

In order to address the concern on counter ions, synthesis of molecule **2** by complexing molecule **S5** with ZnCl_2 has been added and Supplementary Fig. 22 summarizes the similarities between self-assembly properties of molecule **1** and molecule **2**.

Supplementary Fig. 32b has been included which shows the hydrolysis of ATP-1 in the presence of ADP to investigate the mechanistic insights into the enzyme action.

REVIEWERS' COMMENTS:

Reviewer #1 (Remarks to the Author):

This is excellent work. I support publication of the revised manuscript.

Reviewer #2 (Remarks to the Author):

[Ed: reviewer accepts changes made by authors - made no comments to the author but only to the editor]

Reviewer #3 (Remarks to the Author):

I feel the authors have been quite responsive in their revisions and responses to my previous review. Deemphasizing the comparison with actin was a wise move and has made things more clear and less objectionable (at least to me).

A few minor points on the revised manuscript:

1. This addition on lines 27-28 on page 2 relating to ATP hydrolysis in actin seems very out of place and irrelevant for the current work.
2. The bottom paragraph on page 2 - actin does not undergo dynamic instability (this would be microtubules) and actin does **not** require ATP for assembly. It can assemble with ADP and even without any nucleotide (see work from De La Cruz, for example).
3. Page 13 - Fitting seeded growth kinetics is a trivial first-order process since the nucleation phase has been removed from the system and the plot of initial polymerization rate increases linearly with the concentration. Again, papers by Pollard and others make frequent use of this method.

RESPONSE TO REVIEWERS' COMMENTS:

Reviewer #1 (Remarks to the Author):

This is excellent work. I support publication of the revised manuscript.

Response: We thank the reviewer for his support for the publication of the revised manuscript.

Reviewer #3 (Remarks to the Author):

I feel the authors have been quite responsive in their revisions and responses to my previous review. Deemphasizing the comparison with actin was a wise move and has made things more clear and less objectionable (at least to me).

We thank the reviewer for the positive comments and we accordingly believe that decreasing the emphasis on actin polymerization helped improve the quality of our manuscript.

A few minor points on the revised manuscript:

1. This addition on lines 27-28 on page 2 relating to ATP hydrolysis in actin seems very out of place and irrelevant for the current work.

Response: The above mentioned statement was included into the manuscript due to the important work carried out by Voth and coworkers regarding actin assembly and disassembly in their manuscript. And our claim towards a similarities with actin self-assembly depicts the importance of this statement.

*2. The bottom paragraph on page 2 - actin does not undergo dynamic instability (this would be microtubules) and actin does *not* require ATP for assembly. It can assemble with ADP and even without any nucleotide (see work from De La Cruz, for example).*

Response: We thank the reviewer for pointing out these facts and wish to mention that we have modified the text in the manuscript accordingly. We understand that while microtubules undergo dynamic instability, actin filaments exhibit tread milling property which is governed by the rates of hydrolysis of ATP.

We also referred to the work by De La Cruz *et. al.* (*J. Mol. Biol.* **295**, 517-526 (2000)) and realized that they have incorporated specified conditions, under which actin is successfully able to polymerize even in the absence of ATP. While actin filaments do not require ATP to undergo polymerization, we would like to mention that actin polymerizes in Nature by forming ATP-actin monomers which undergo nucleation-elongation process. Though we understand the concerns of the reviewer and have modified the text in manuscript accordingly.

3. Page 13 - Fitting seeded growth kinetics is a trivial first-order process since the nucleation phase has been removed from the system and the plot of initial polymerization rate

increases linearly with the concentration. Again, papers by Pollard and others make frequent use of this method.

Response: We agree with the reviewer's comments that fitting seeded growth kinetics is a trivial first order process in the case of actin polymerization. But the fact that our synthetically designed molecule **1** undergoes a similar process and follows a similar trend on fitting the seeded growth kinetics is important to mention in the manuscript. This is a unique result for a fuel-driven seeded polymerization process of a synthetic system (Figure 4e).